

# Ecological stoichiometry of plant leaves, litter and soils in a secondary forest on China's Loess Plateau

Zongfei Wang[1,2] and  Fenli Zheng[1,3]

[1] State Key Laboratory of Soil Erosion and Dryland Farming on the Loess Plateau, Institute of Soil and Water Conservation, Chinese Academy of Sciences and Ministry of Water Resources, Yangling, China
[2] University of Chinese Academy of Sciences, Beijing, China
[3] Institute of Soil and Water Conservation, Northwest A&F University, Yangling, China

## ABSTRACT

Ecological stoichiometry can reveal nutrient cycles in soil and plant ecosystems and their interactions. However, the ecological stoichiometry characteristics of leaf-litter-soil system of dominant grasses, shrubs and trees are still unclear as are their intrinsic relationship during vegetation restoration. This study selected three dominant plant types of grasses (*Imperata cylindrica* (*I. cylindrica*) and *Artemisiasacrorum* (*A.sacrorum*)), shrubs (*Sophora viciifolia* (*S. viciifolia*) and *Hippophae rhamnoides* (*H. rhamnoides*)) and trees (*Quercus liaotungensis* (*Q. liaotungensis*) and *Betula platyphylla* (*B. platyphylla*)) in secondary forest areas of the Chinese Loess Plateau to investigate ecological stoichiometric characteristics and their intrinsic relationships in leaf-litter-soil systems. The results indicated that N concentration and N:P ratios in leaf and litter were highest in shrubland; leaf P concentration in grassland was highest and litter in forestland had the highest P concentration. Soil C, N and P concentrations were highest in forestland ($P < 0.05$) and declined with soil depth. Based on the theory that leaf N:P ratio indicates nutritional limitation of plant growth, this study concluded that grass and shrub growth was limited by N and P element, respectively, and forest growth was limited by both of N and P elements. The relationships between the N concentration in soil, leaf and litter was not significant ($P > 0.5$), but the soil P concentration was significantly correlated with litter P concentration ($P < 0.05$). These finding enhance understanding of nutrient limitations in different plant communities during vegetation restoration and provide insights for better management of vegetation restoration.

# INTRODUCTION

Soil erosion remains a major global environmental problem, accelerating soil nutrient losses and ecosystem degradation (*Luque et al., 2013*). Soil nutrient losses greatly decreased soil quality (*Liu & Dang, 1993*), which seriously threatens the stability of ecosystems. Vegetation restoration is a powerful approach for ecological restoration of degraded lands, as it can control soil erosion and improve ecosystem functions and services (*Godefroid et al., 2003*; *Zheng, 2006*; *Jiao et al., 2012*; *Sauer et al., 2012*; *Zhao et al., 2015*;

Corresponding author
Fenli Zheng, flzh@ms.iswc.ac.cn

*Bienes et al., 2016*). Vegetation restoration areas currently cover approximately 0.20 billion ha worldwide and are being planted at a rate of 4.5 million ha per year (*Zhao et al., 2015*). Over time, vegetation restoration can improve soil quality (*Fu et al., 2010*), and accelerate N and P cycling in plants and soils (*Lü et al., 2012*). Vegetation restoration has resulted in species replacement, which has changed the structure of the community and species diversity (*Wang et al., 2011*) and form a diverse ecosystem of trees, shrubs, and herbs, which results in changes in nutrients distribution in leaves, litter and soil (*Parfitt, Yeates & Ross, 2005*; *Hobbie et al., 2006*; *John et al., 2007*; *Jiao et al., 2013*; *Zhao et al., 2017*). Several plant communities show significant differences in nutrient allocation due to different plant species throughout vegetation restoration (*Warren & Zou, 2002*; *Schreeg et al., 2014*; *Deng et al., 2016*). Therefore, it is necessary to quality nutrient characteristics in the leaf-litter-soil system of dominant grasses, shrubs and trees, as well as their intrinsic relationships during vegetation restoration.

Ecological stoichiometry describes the balance of energy and multiple chemical elements in ecosystems (*Elser et al., 2000*), and has already become a method for studying the stability and N/P limitations of degraded ecosystems (*Güsewell, 2004*; *Han et al., 2005*). Ecological stoichiometry is also an effective tool to study the interactions between soils and plant, and their nutrient cycles (*Elser, 2006*). C, N, and P cycles account for the transfer of nutrients between plant and soil. C is a key building block of structural substances, supplying approximately 50% of the dry biomass, whereas N and P are the major limiting elements of natural terrestrial ecosystems and both play important roles in several physiological and metabolic processes. These three nutrients elements interact with each other, and both N and P affect carbon fixation (*Han et al., 2005*). The notion that leaf N:P ratio can be used to identify nutrient limitations for plant growth has been widely confirmed in various plant communities (*Koerselman & Meuleman, 1996*; *Schreeg et al., 2014*). The N:P ratio of plant leaves can be used to characterize the productivity of terrestrial ecosystems, and it can also indicate which elements of the plant are limited, but this relationship can change with changes in the environment (*Güsewell, 2004*). Thus, it provides a scientific basis for the rational allocation of vegetation to investigate nutrient limitation of N:P ratio in the process of vegetation restoration.

In the plant-soil ecosystem, litter serves as a main carrier of nutrients and links plants and soil (*Agren & Bosatta, 1998*). The litter layer provides storage for ecosystems nutrients and acts as a hub for material exchange between soils and plants, and it is a natural source of soil fertility (*Agren et al., 2013*). Nutrient supply in soil, plant growth demand, and litter return to soil are nominally independent factors, but they also interact with each other, which leads to the complex relationship among nutrient concentrations in the plant-litter-soil systems (*Agren & Bosatta, 1998*). Ecological stoichiometry provides an effective approach for observing these relationship between nutrients in the plant-litter-soil systems and their characteristics in ecological processes (*Elser et al., 2000*). Thus, it is of theoretical and practical significance to analyze the ecological stoichiometric characteristics of leaf-litter-soil systems during vegetation restoration.

Due to its steep topography and erodible soil, coupled with long-term human activity, the ecological environment of the Loess Plateau is extremely fragile, and has become

one of the most severely eroded areas of China (*Jiao et al., 2012*; *Zhao et al., 2015*). In past centuries, the majority of forestlands were destroyed to satisfy the food needs of the growing population, which resulted in severe soil erosion and land degradation. The Grain to Green Program (GTGP) was implemented to control soil erosion and improve ecosystem degradation, with a main goal of converting low-yield steep-slope croplands into permanent vegetation cover (*Jiao et al., 2012*; *Zhao et al., 2015*). Vegetation restoration generated a diverse flora and reduced soil erosion, raising interest in the characterization of this recovering ecosystem. For example, *An & Shangguan (2010)* and *Chai et al. (2015)* studied leaf stoichiometric traits and concluded that the growth of vegetation was N-limited at each secondary successional stage, according to the leaf N:P threshold. *Ai et al. (2017)* observed that the slope aspect had various effects on plant and soil C:N:P stoichiometry. Variations in vegetation types influenced soil C:N:P ratios, which were higher in afforested lands than in slope croplands (*Zhao et al., 2015*; *Deng et al., 2016*; *Zhao et al., 2017*). *Jiao et al. (2013)* studied soil stoichiometry during vegetation successional changes and reported that soil N:P ratio increased with the vegetation restoration year. It was even reported that forest age had a significant effect on C, N, P and K concentrations and their ratios in plant tissues and soil (*Li et al., 2013*). Most previous studies addressed the stoichiometric characteristics of soil system and vegetation communities, including forests and grasslands, as well as litter individually or in both. However, the ecological stoichiometry of the plant-litter-soil system as a whole has been rarely described (*Zeng et al., 2017*; *Cao & Chen, 2017*), and the effects of dominant plant communities (tree, shrub, grass) during vegetation restoration on this ecological stoichiometry remains poorly understood. This will provide a better understanding of nutrient limitation in different plant communities during vegetation restoration and improve ecosystem management. In addition, the majority of previous studies have focused on topsoil (*Jiao et al., 2013*; *Li et al., 2013*; *Zeng et al., 2016*; *Zeng et al., 2017*), there is little information on stoichiometry change with the soil profile (*Zhao et al., 2015*; *Deng et al., 2016*). Due to the depth of thick loess on the Loess Plateau, the majority plant roots are distributed within the top 100 cm. Therefore, it is important to investigate change of the stoichiometry of C, N and P with soil profile depth.

Vegetation succession starts from annual grass, then to perennial grass, shrub, forest after farmlands are abandoned. So, three dominant plant communities of grasses (*Imperata cylindrica* and *Artemisia sacrorum*), shrubs (*Sophora viciifolia* and *Hippophae rhamnoides*) and trees (*Quercus liaotungensis* and *Betula platyphylla*) were selected in the Ziwuling secondary area of the Loess Plateau to investigate ecological stoichiometry in the plant-litter-soil system and their intrinsic relationships. The specific objectives of this study were to (1) determine leaf and litter C, N and P concentrations and their ecological stoichiometry characteristics in six dominant plant species; (2) investigate distributions of soil C, N, and P concentrations and ecological stoichiometry characteristics in soil profile; (3) examine the relationships of ecological stoichiometry in leaf-litter-soil system (C, N, and P); and (4) assess the limiting nutrient element for plant growth in the six plant species. The effort will provide information about ecological stoichiometry and theoretical support for enhancing vegetation and ecosystem restoration on the Loess Plateau.

## MATERIAL AND METHODS

### Study site description

This study site is located at Fuxian County, Shanxi Province, China (35°5.4′N, 109°8.9′E), in the center of Loess Plateau, south of the Yan'an city. The topography and landform belong to loess hilly-gully region with elevation ranging from 920 to 1,683 m (*Zheng, 2006*). The mean annual temperature ranges from 6 to 10 °C and mean annual precipitations is between 600 to 700 mm. The soil is mainly composed of loess, which can be classified as a Calcic Cambisol (*USDA NRCS, 1999*). The soil texture was 28.3% sand (>50 μm), 58.1% silt (50–2 μm) and 13.6% clay (<2 μm). Vegetation of the Loess Plateau was almost completely removed more than 100 years ago, and soil loss was 8,000 to 10,000 t km$^{-2}$ yr$^{-1}$ (*Zheng et al., 1997*; *Kang et al., 2014*). In 1866–1870, the inner war happened in this region (*Zhang & Tang, 1992*; *Tang et al., 1993*; *Zheng et al., 1997*) and as population moved out, and then secondary succession of vegetation began. Currently, forest canopy closure is more than 0.6 and dominant species for tree are *Quercus liaotungensis* (climax forest community) and *Betula platyphylla* (early forest community); dominant species for shrub are *Sophora viciifolia* and *Hippophae rhamnoides*, both does not concur in same places; and main grass species are *Imperata cylindrica* and *Artemisia sacrorum* (*Zheng, 2006*). The distribution area of the above mentioned six dominant species occupies more than 70% of total area in the study site.

### Soil and plant sampling

According to our field investigation, there are 38 species in the study site, including 18 artificial species and 20 natural species, which cover five tree species, six shrub species, nine grass species. Moreover, the six tree, shrub, and grass species, i.e., *Quercus liaotungensis* and *Betula platyphylla* (forest communities), *Sophora viciifolia* and *Hippophae rhamnoides* (shrub communities) and *Imperata cylindrica* and *Artemisia sacrorum* (grass communities) are dominant species and their distribution area occupies more than 70% of total area in the study site. Other studies also reported that these six species are dominant species of natural vegetation succession (*Zheng, 2006*; *Wang, Shao & Shangguan, 2010*; *Zhang & Shangguan, 2016*). Thus, these six species have been selected to investigate to ecological stoichiometry of plant leaf, litter and soil in a secondary forest on China's Loess Plateau. For each dominant species, three experimental sites (three replications) with a similar site condition including slope position (slope length, gradient and aspect), soil type and altitude were set up to collect samples. In addition, the distance within all experimental sites was within approximately 3 km, which reduced impacts of previous site condition. Plant leaves and soil samples were collected in late July 2016 when plants were in a vigorous growth period, and litter samples on the soil surface consisting of leaf fall over multiple years that were not decomposed were obtained in late October 2016. Table 1 shows the characteristics of these three plant types.

Two plots with 10 × 10 m size were established in each experimental site of forest type, and the plots sizes for shrub and grass types were 5 × 5 m and 1 × 1 m, respectively. Ten to twenty complete expanded living and sun-exposed leaves were randomly collected from five to ten healthy individual plants per plot from shrubs or trees, and a total of 80 to 100

**Table 1  Characteristics of the three plant types.**

| Vegetation types | Dominant plant species | Abbreviation | Accompanying plant species | Altitude (m) | Coverage (%) | Slope degree (°) | Slope aspect |
|---|---|---|---|---|---|---|---|
| Forest | *Quercus liaotungensis* | *Q. liaotungensis* | *Carex lanceolata* | 1,355 | 60 | 21–25 | WS260° |
| | *Betula platyphylla* | *B. platyphylla* | | 1,133 | 80 | 17–20 | WS120° |
| Shrub | *Sophora viciifolia* | *S. viciifolia* | *Stipa bungeana* | 1,280 | 55 | 15–20 | WS255° |
| | *Hippophae rhamnoides* | *H. rhamnoides* | *Buddleja alternifolia* | 1,332 | 75 | 15–17 | WS45° |
| Grass | *Imperata cylindrica* | *I. cylindrica* | *Artemisia giraldii* | 1,310 | 70 | 10–12 | WS259° |
| | *Artemisia sacrorum* | *A. sacrorum* | *Themeda japonica* | 1,336 | 75 | 15–20 | WS220° |

leave samples were collected. For each grass plot, all stems and leaves were completely cut from three 0.25 m$^2$ sampling areas. Leaves from each plot were evenly mixed and then put into a paper bag. Litter samples were collected along the diagonal lines of three 1 × 1 m squares per plot, and mixed and stored in paper bags. All samples of leaves and litter were carried back to the indoor laboratory for analysis.

The total of 256 soil samples from a 100 cm-depth profile were collected using a 5-cm diameter to collect soil samples along an S-shaped line in each plot. Before each soil sample was collected, soil sampler was sterilized with ethanol to avoid cross-infection. Moreover, the 100 cm soil profile was divided into six layers (0–10, 10–20, 20–40, 40–60, 60–80, 80–100 cm), and soil samples from each layer were obtained from five points. The five soil samples of each layer were mixed evenly and stored in plastic bags, and then all soil samples (6 plant species ×3 experimental sites ×2 sample plots ×6 soil sample layers) were transported to the indoor laboratory.

## Sample analysis

Leaf and litter samples were oven dried at 70 °C for at least 48 h or more to reach a constant mass level, and then weighed. Dried plant samples were ground to a fine powder using a plant-sample mill (1,093 Sample Mill, Foss, Sweden). Soil samples were air-dried and sieved using a 0.25 mm mesh. To determine C concentration in plant and soil, the Walkley-Black modified acid-dichromate FeSO$_4$ titration method was used (*Bao, 2000*), and the Kjeldahl method (KJELTE2300, Sweden) was applied to measure the total N concentration in plant and soil. The total P concentration in plant was measured by using a Spectrophotometer UV-2300 (Techcomp Com, Shanghai, China) after digestion with H$_2$SO$_4$ and H$_2$O$_2$, and the total P concentration in soil was determined by a spectrophotometer after wet digestion with H$_2$SO$_4$ and HClO$_4$ (*Bao, 2000*). Leaf, litter and soil C, N, P concentrations were expressed as g/kg on dry weight basis. The C:N:P ratios in leaves, litter and soil were computed as mass ratios.

## Statistical analysis

All data are presented as mean ± standard errors and tested for normality of distributions and homogeneity of variances before analysis. A one-way analysis of variance (ANOVA) was used to analyze the effects of the plant type on nutrients and stoichiometric characteristics in leaf, litter and soil. Two-way ANOVAs were computed to analyze the effects of plant type, soil depth and their interactions on soil C, N and P concentrations and their stoichiometry.

**Table 2** Nutrient concentrations and characteristics of ecological stoichiometry in leaves of the three plant types.

| Vegetation types | Plant species | C /(g/kg) | N /(g/kg) | P /(g/kg) | C:N | C:P | N:P |
|---|---|---|---|---|---|---|---|
| Forest | *Q. liaotungensis* | 505 ± 11.5Bb | 18.7 ± 0.98Ab | 1.30 ± 0.03Bc | 27.1 ± 0.86Ac | 388 ± 6.67Aa | 14.3 ± 0.55Ac |
| | *B. platyphylla* | 522 ± 11.4Aa | 18.1 ± 1.00Ab | 1.40 ± 0.03Ab | 29.0 ± 2.04Ab | 373 ± 24.0Ab | 12.9 ± 0.15Bd |
| Shrub | *S. viciifolia* | 499 ± 9.62Ab | 28.9 ± 0.83Aa | 1.28 ± 0.03Bc | 17.3 ± 0.23Ad | 390 ± 10.8Aa | 22.6 ± 0.82Aa |
| | *H. rhamnoides* | 502 ± 12.2Ab | 29.8 ± 1.24Aa | 1.42 ± 0.03Ab | 16.9 ± 0.77Ad | 354 ± 12.3Bb | 21.0 ± 0.97Ba |
| Grass | *I. cylindrica* | 491 ± 5.33Ab | 10.4 ± 0.32Bc | 1.70 ± 0.03Aa | 47.3 ± 1.73Aa | 289 ± 8.13Ac | 6.12 ± 0.21Bf |
| | *A. sacrorum* | 475 ± 9.97Bc | 17.9 ± 0.38Ab | 1.80 ± 0.10Aa | 26.6 ± 0.78Bc | 264 ± 12.0Ac | 9.93 ± 0.71Ae |

**Notes.**

Bars indicate the standard errors ($n = 6$). The lowercase letters above the bars indicate significant differences in leaf at different plant types and the capital letters represent significant differences in leaf at the same plant types of different species ($P < 0.05$).

The linear regression analysis was used to test the relationship between C, N and P concentrations in leaf, litter and soil. Pearson correlation was used to assess relationship between leaf, litter and soil stoichiometric characteristics. Differences were considered significant with a $P < 0.05$. All statistical analyses were determined with SPSS 19.0 software (SPSS, Inc., Chicago, IL, USA).

# RESULTS

## Leaf and litter nutrients and ecological stoichiometry in dominant plant communities

The leaf C, N and P concentrations were different among plant communities (Table 2). The C concentration in leaf varied from 475 (grass) to 522 g/kg (forest), and was highest in *B. platyphy* lla and lowest in *A. sacrorum*. The leaf N concentration was 29.8 g/kg in shrub, and was significantly greater than in forest and grass ($P < 0.05$), while the leaf P concentration with 1.80 g/kg was highest in grass. The leaf C:N ratio varied from 16.9 (shrub) to 47.3 (grass), and was highest in *I. cylindrica* and lowest in *H. rhamnoides*. The leaf C:P ratio was significantly higher in *Q. liaotungensis* and *S. viciifolia* than other species ($P < 0.05$). The leaf N:P ratio varied from 6.12 (grass) to 22.6 (shrub) and was significantly higher in shrub than in grass and forest ($P < 0.05$).

The C, N and P concentrations in litter were significantly affected by plant types (Table 3). The litter C concentration varied from 360 (shrub) to 413 (forest), and was significantly higher in forest than in grass and shrub ($P < 0.05$). N concentrations showed a similar pattern between litter and leaf, and were significantly highest in shrub ($P < 0.05$). The litter P concentration varied from 0.51 (grass) to 0.97 g/kg (forest) and was highest in *B. platyphy* lla and lowest in *I. cylindrica*. The litter C:N and C:P ratios in grass were 52.9 and 735, respectively, and were significantly higher than in forest and shrub ($P < 0.05$). The litter N:P ratio varied from 12.5 (forest) to 24.2 (shrub), and was highest in *H. rhamnoides* and lowest in *B. platyphy* lla ($P < 0.05$).

**Table 3   Nutrient concentrations and characteristics of ecological stoichiometry in litter of the three plant types.**

| Vegetation types | Plant Species | C /(g/kg) | N /(g/kg) | P /(g/kg) | C:N | C:P | N:P |
|---|---|---|---|---|---|---|---|
| Forest | *Q. liaotungensis* | 398 ± 11.5Aab | 13.8 ± 0.82Ab | 0.92 ± 0.04Aa | 29.1 ± 2.53Bc | 433 ± 29.1Ad | 14.9 ± 0.38Ac |
| | *B. platyphylla* | 413 ± 15.4Aa | 12.2 ± 1.91Ac | 0.97 ± 0.10Aa | 34.6 ± 4.26Ab | 431 ± 35.3Ad | 12.5 ± 0.66Bd |
| Shrub | *S. viciifolia* | 360 ± 16.7Ac | 17.5 ± 1.12Aa | 0.75 ± 0.06Ab | 20.7 ± 1.48Ad | 486 ± 51.0Ac | 23.5 ± 1.52Aa |
| | *H. rhamnoides* | 360 ± 26.2Ac | 17.7 ± 0.56Aa | 0.74 ± 0.06Ab | 20.4 ± 2.00Ad | 489 ± 29.2Abc | 24.2 ± 2.27Aa |
| Grass | *I. cylindrica* | 375 ± 12.6Abc | 7.12 ± 0.48Bd | 0.51 ± 0.03Bc | 52.9 ± 3.84Aa | 735 ± 42.7Aa | 14.0 ± 1.61Bc |
| | *A. sacrorum* | 395 ± 17.4Aab | 12.0 ± 0.69Ac | 0.73 ± 0.04Ab | 33.0 ± 2.43Bb | 543 ± 42.2Bb | 16.5 ± 0.53Ab |

**Notes.**

Bars indicate the standard errors ($n = 6$). The lowercase letters above the bars indicate significant differences in litter at different plant types and the capital letters represent significant differences in litter at the same plant types of different species ($P < 0.05$).

## Soil nutrients and ecological stoichiometry in dominant plant communities and soil depths

Plant type and soil depth had significant effects on soil nutrients and their C:N:P ratios (Table 4). Soil C and N concentrations in forestland were greater than in shrubland and grassland at all soil depths and both were highest in *Q. liaotungensis* and lowest in *A. sacrorum* ($P < 0.05$). Soil P concentration in shrubland was lower than in grassland and forestland at every soil depth ($P < 0.05$), and there were no differences in *B. platyphylla*, *Q. liaotungensis* and *A. sacrorum* at 20–100 cm soil depths. Soil C:N ratio in forestland was significantly higher than in shrubland and grassland at both 0-10 and 10–20 cm soil depths ($P < 0.05$), but there were no significant differences at 20–100 cm soil depths ($P > 0.05$). Soil C:P and N:P ratios in forestland was significantly higher than in shrubland and grassland at both 0-10 and 10–20 cm soil depths ($P < 0.05$), but both were highest in shrubland at 20–100 cm soil depths ($P < 0.05$).

Soil depth is a driving factor for soil nutrient concentrations and their ratios (Table 4 and Fig. 1). Soil C and N concentrations significantly decreased with soil sampling depth. Soil C and N concentrations decreased markedly from 10 to 40 cm of soil depth, and then slightly decreased from 40 to 100 cm. Soil P concentration tended to stable with the soil sampling depth. Soil C:N ratio fluctuated with depth, and soil C:P and N:P ratios had the same trend along the soil sampling depth and decreased markedly from 10 to 40 cm of soil depth, and then slightly decreased from 40 to 100 cm.

The results of the Two-way ANOVA analysis indicated that both plant type and soil depth significantly affected the soil C, N and P concentrations and their stoichiometry (C:N, C:P and N:P ratios). The interactions between plant type and soil depth significantly affected the soil C and N concentrations and C:N, C:P and N:P ratios but not soil P concentration (Table 5).

## Relationships between C, N and P concentrations and their characteristics of ecological stoichiometry among leaf, litter and soil

There were significant correlations between leaf and litter for both N and P concentrations in three plant community types ($P < 0.05$) (Figs. 2B, 2C). The relationships between the plant C concentration and soil C concentration were significant in two soil layers (0–10

**Table 4 Profile distribution of soil nutrient concentrations and characteristics of ecological stoichiometry at different community types.**

| Vegetation community | Soil layer (cm) | C/(g/kg) | N/(g/kg) | P/(g/kg) | C:N | C:P | N:P |
|---|---|---|---|---|---|---|---|
| Q. liaotungensis | 0–10 | 21.9 ± 0.69Aa | 1.83 ± 0.13Aa | 0.70 ± 0.02Ab | 12.0 ± 0.53Aa | 31.5 ± 1.49Aa | 2.64 ± 0.22Aa |
| | 10–20 | 12.6 ± 0.59Ba | 0.99 ± 0.02Ba | 0.66 ± 0.03Bab | 12.7 ± 0.56Aa | 19.1 ± 1.48Ba | 1.50 ± 0.06Bb |
| | 20–40 | 5.51 ± 0.41Ca | 0.61 ± 0.06Ca | 0.64 ± 0.02BCa | 9.13 ± 0.6Ca | 8.65 ± 0.69Cb | 0.91 ± 0.09Cb |
| | 40–60 | 4.25 ± 0.24Dab | 0.49 ± 0.04Da | 0.62 ± 0.02Cb | 8.72 ± 0.32Cc | 6.88 ± 0.27Dcd | 0.79 ± 0.06Dab |
| | 60–80 | 4.04 ± 0.25 Dab | 0.43 ± 0.03Da | 0.63 ± 0.02Ca | 9.49 ± 0.29BCa | 6.40 ± 0.32Dbc | 0.67 ± 0.03DEbc |
| | 80–100 | 3.94 ± 0.25 Da | 0.40 ± 0.06Da | 0.63 ± 0.02Ca | 9.55 ± 0.91Ba | 6.28 ± 0.18Dbc | 0.64 ± 0.07Ebc |
| B. platyphylla | 0–10 | 19.0 ± 1.65Ab | 1.67 ± 0.70Ab | 0.74 ± 0.05Aa | 11.3 ± 0.57Aab | 25.7 ± 1.04Ab | 2.27 ± 0.07Ab |
| | 10–20 | 11.3 ± 0.65Bb | 0.97 ± 0.06Ba | 0.69 ± 0.03ABa | 11.7 ± 0.31Ab | 16.2 ± 0.42Bb | 1.39 ± 0.05Bc |
| | 20–40 | 5.46 ± 0.41Ca | 0.59 ± 0.04Ca | 0.66 ± 0.03BCa | 9.28 ± 1.28Ca | 8.32 ± 0.64Cb | 0.91 ± 0.07Cb |
| | 40–60 | 4.48 ± 0.53CDa | 0.47 ± 0.05Dab | 0.65 ± 0.03BCa | 9.45 ± 0.57Cabc | 6.95 ± 0.95Dabc | 0.74 ± 0.09Dc |
| | 60–80 | 4.22 ± 0.22Da | 0.40 ± 0.03Eab | 0.64 ± 0.06BCa | 10.6 ± 1.03ABa | 6.71 ± 1.02Dbc | 0.63 ± 0.07Ebc |
| | 80–100 | 3.69 ± 0.32Dab | 0.37 ± 0.03Eab | 0.62 ± 0.05Ca | 9.93 ± 0.94BCa | 5.98 ± 0.76Dbc | 0.61 ± 0.08Ec |
| S. viciifolia | 0–10 | 15.4 ± 0.94Ac | 1.43 ± 0.0.9Ac | 0.60 ± 0.03Ac | 10.8 ± 1.17Ab | 25.8 ± 1.32Ab | 2.41 ± 0.19Ab |
| | 10–20 | 9.58 ± 0.69Bc | 0.89 ± 0.02Bb | 0.58 ± 0.01Ac | 10.8 ± 0.92ABc | 16.7 ± 0.96Bb | 1.55 ± 0.05Bb |
| | 20–40 | 5.27 ± 0.41Cab | 0.57 ± 0.03Ca | 0.58 ± 0.03Ab | 9.26 ± 0.35Ca | 9.13 ± 0.39Cb | 0.99 ± 0.04Cb |
| | 40–60 | 3.98 ± 0.38Dc | 0.43 ± 0.08Dabc | 0.54 ± 0.00Bc | 9.48 ± 1.16BCabc | 7.38 ± 0.70Dbc | 0.80 ± 0.15Dab |
| | 60–80 | 3.52 ± 0.29Dc | 0.37 ± 0.06Deb | 0.52 ± 0.00Bb | 9.79 ± 1.08ABCa | 6.76 ± 0.38Dbc | 0.70 ± 0.11Db |
| | 80–100 | 3.40 ± 0.20Dc | 0.33 ± 0.03Ec | 0.52 ± 0.01Bb | 10.3 ± 0.51ABCa | 6.55 ± 0.49Db | 0.64 ± 0.05Dbc |
| H. rhamnoides | 0–10 | 11.9 ± 1.12Ad | 1.11 ± 0.08Ae | 0.49 ± 0.01Ad | 10.8 ± 0.81Ab | 24.1 ± 2.07Ab | 2.25 ± 0.14Ab |
| | 10–20 | 8.11 ± 0.56Bd | 0.85 ± 0.05Bb | 0.45 ± 0.03Bd | 9.55 ± 0.82ABd | 18.1 ± 1.94Ba | 1.90 ± 0.16Ba |
| | 20–40 | 4.45 ± 0.19Cc | 0.48 ± 0.01Cb | 0.42 ± 0.0.01Cc | 9.31 ± 0.39Ba | 10.7 ± 0.67Ca | 1.15 ± 0.05Ca |
| | 40–60 | 3.80 ± 0.27CDc | 0.38 ± 0.03Dc | 0.42 ± 0.01Cd | 10.1 ± 1.08abAB | 9.08 ± 0.66CDa | 0.91 ± 0.08Da |
| | 60–80 | 3.61 ± 0.08Dc | 0.35 ± 0.03Db | 0.42 ± 0.01Cc | 10.3 ± 1.06ABa | 8.61 ± 0.64Da | 0.84 ± 0.08Da |
| | 80–100 | 3.40 ± 0.22Dc | 0.36 ± 0.04Dab | 0.43 ± 0.02Cc | 9.56 ± 1.04ABa | 8.01 ± 0.29Da | 0.85 ± 0.09Da |
| I. cylindrica | 0–10 | 13.2 ± 1.05Ad | 1.27 ± 0.06Ad | 0.64 ± 0.02Ac | 10.4 ± 0.43Abc | 20.7 ± 1.01Ac | 1.99 ± 0.03Ac |
| | 10–20 | 8.93 ± 0.72Bc | 0.86 ± 0.07Bb | 0.60 ± 0.03Bc | 10.5 ± 0.26Ac | 14.8 ± 0.47Bc | 1.42 ± 0.05Bc |
| | 20–40 | 5.03 ± 0.53Cab | 0.58 ± 0.01Ca | 0.58 ± 0.02Bb | 8.72 ± 0.90Bab | 8.69 ± 1.02Cb | 1.00 ± 0.05Cb |
| | 40–60 | 4.20 ± 0.27CDab | 0.41 ± 0.06Db | 0.54 ± 0.01Cc | 10.5 ± 1.23Aa | 7.74 ± 0.56CDb | 0.75 ± 0.10Dc |
| | 60–80 | 3.70 ± 0.37Dbc | 0.37 ± 0.03Dbc | 0.52 ± 0.02CDb | 10.1 ± 1.01Aa | 7.09 ± 0.61Db | 0.71 ± 0.07Db |
| | 80–100 | 3.68 ± 0.42Dab | 0.36 ± 0.05Dab | 0.50 ± 0.01Db | 10.4 ± 1.41Aa | 7.34 ± 0.76Da | 0.72 ± 0.10Db |
| A. sacrorum | 0–10 | 8.96 ± 0.29Ae | 0.92 ± 0.02Af | 0.68 ± 0.03Ab | 9.74 ± 0.38ABc | 13.2 ± 0.67Ad | 1.36 ± 0.11Ad |
| | 10–20 | 7.31 ± 0.28Bd | 0.78 ± 0.05Bc | 0.65 ± 0.03Bb | 9.46 ± 0.40Bd | 11.2 ± 0.34Bd | 1.19 ± 0.05Bd |
| | 20–40 | 4.74 ± 0.37Cbc | 0.59 ± 0.06Ca | 0.64 ± 0.02BCa | 8.02 ± 0.31Cb | 7.45 ± 0.43Cc | 0.93 ± 0.07Cb |
| | 40–60 | 4.06 ± 0.19Dab | 0.45 ± 0.04Dabc | 0.62 ± 0.01CDab | 9.17 ± 0.48Bbc | 6.52 ± 0.19Dd | 0..71 ± 0.06Dc |
| | 60–80 | 3.60 ± 0.36Ec | 0.37 ± 0.02Eb | 0.61 ± 0.02CDa | 9.88 ± 1.32ABa | 5.90 ± 0.46Ec | 0.60 ± 04Ec |
| | 80–100 | 3.49 ± 0.27Ec | 0.34 ± 0.03Ec | 0.61 ± 0. 02Da | 10.4 ± 0.54Aa | 5.76 ± 0.33Ec | 0.55 ± 0.03Ec |

**Notes.**
Bars indicates the standard errors ($n = 6$). The lowercase letters above the bars indicate significant differences in different plant species at the same four soil layers, and the capital letters represent significant differences in different soil layers at the same plant species ($P < 0.05$).

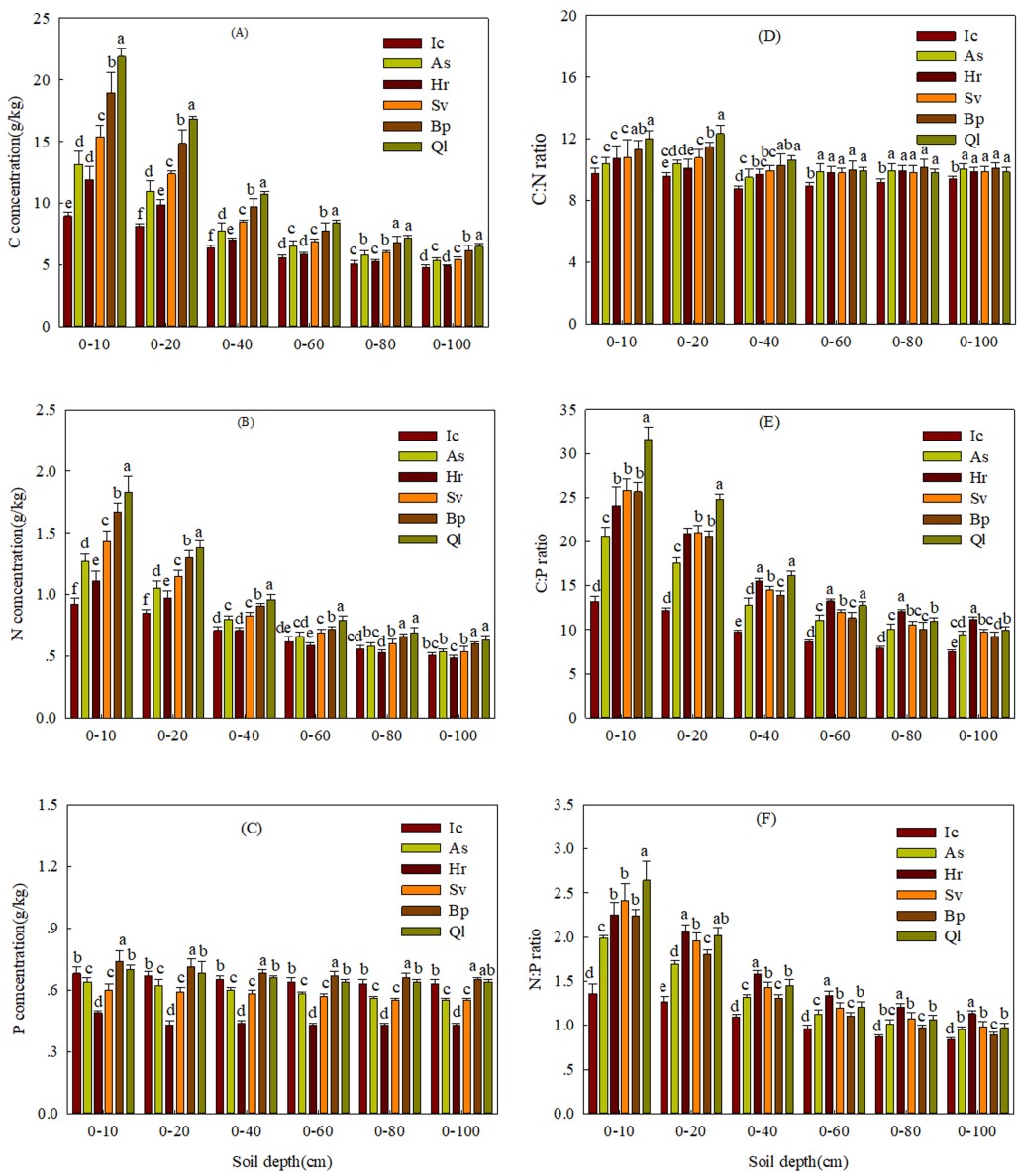

**Figure 1 Concentrations of soil C, N, P and their ecological stoichiometry in the different sampling soil layers of the different plant species.** (A) Soil C concentration in the different sampling soil layers of the different plant species. (B) Soil N concentration in the different sampling soil layers of the different plant species. (C) Soil P concentration in the different sampling soil layers of the different plant species. (D) Soil C:N ratio in the different sampling soil layers of the different plant species. (E) Soil C:P ratio in the different sampling soil layers of the different plant species. (F) Soil N:P ratio in the different sampling soil layers of the different plant species. Bars indicates the standard errors ($n = 6$). The lowercase letters above the bars indicate significant differences in different plant species at the same soil layers, and the capital letters represent significant differences in different soil layers at the same plant species ($P < 0.05$). Ic and As represent *I. cylindrica* and *A. sacrorum*, respectively; Hr and Sv represent *H. rhamnoides* and *S. viciifolia*, respectively; Bp and Ql represent *B. platyphylla* and *Q. liaotungensis*, respectively.

**Table 5  Correlations among ecological stoichiometry in leaf, litter and soil at 0–10, 0–20 and 0–100 cm soil depth.**

| Factor | $F(P)$value | | | | | |
|---|---|---|---|---|---|---|
| | C | N | P | C:N | C:P | N:P |
| Plant type | 153 (<0.0001[b]) | 81.4 (<0.0001) | 315 (<0.0001) | 4.90 (0.0003) | 125 (<0.0001) | 72.1 (<0.0001) |
| Soil depth | 1999 (<0.0001) | 1701 (<0.0001) | 51.6 (<0.0001) | 22.4 (<0.0001) | 1859 (<0.0001) | 1226 (<0.0001) |
| Plant type; Soil depth[a] | 54.1 (<0.0001) | 30.4 (<0.0001) | 0.35 (0.0663) | 3.54 (<0.0001) | 40.1 (<0.0001) | 18.8 (<0.0001) |

**Notes.**
[a]Indicates the interaction between plant type and soil depth.
[b]parentheses is $P$ value.

and 0–20) and the profile (0–100 cm) ($P < 0.05$) (Figs. 3A, 3B, 3C and Figs. 4A, 4B, 4C), while there were no significant correlation between plant N concentration and soil N concentration (Figs. 3D, 3E, 3F and Figs. 4D, 4E, 4F). In the three plant community types, there were no significant correlations between leaf P concentration and soil P concentration (Figs. 3G, 3H, 3I), but the soil P concentration was significant correlated with litter P concentration in 0–10 cm soil depth ($P < 0.05$) (Fig. 3G).

For the three plant community types, leaf C:N and N:P ratios were positively correlated with litter C:N and N:P ratios, respectively ($P < 0.05$) (Figs. 2D, 2F), while leaf C:P ratio was negatively correlated with litter C:P ratio ($P < 0.05$) (Fig. 2E). Leaf C:P had a positive correlation with soil C:P ratio at the 0–10 cm soil layer and over 0–100 cm soil profile ($P < 0.05$) (Table 6), Leaf N:P ratio had a positive correlation with soil N:P ratio at two soil layers (0–10 and 0–20 cm) and the profile (0–100 cm)($P < 0.05$) (Table 6); and there was significant correlation between leaf and soil C:N ratio at the 0–10 and 0–20 cm soil layers ($P < 0.05$) (Table 6). At the 0–10 cm soil layer, there was a significant correlation between litter and soil C:N ratio ($P < 0.05$) (Table 6), and litter C:P ratios were negatively correlated with C:P ratios at two soil layers (0–10 and 0–20 cm) and the profile (0–100 cm), and only in the profile (0–100 cm) did litter N:P ratio have a positive correlation with soil N:P ratio ($P < 0.05$) (Table 6).

## DISCUSSION

### Impacts of dominant plant communities on leaf and litter nutrients and ecological stoichiometry

As a key subsystem, plants have a vital function in governing the stability of terrestrial ecosystem. C, N and P are essential nutrients for plant (*Han et al., 2005*; *John et al., 2007*) and their interaction regulate plant growth (*Güsewell, 2004*). Litter is one main way for nutrients to return to the soil and is an important part of the forest ecosystem. The decomposition of plant litter replenishes soil nutrients to provide conditions for the adjustment and demand of the plant nutrients (*Agren & Bosatta, 1998*). There are differences in the types, quantity and utilization efficiency of absorbed nutrients in different plants types. In this study, the results indicated that leaf C, N and P concentrations differed across plant communities. The reason is that different plant communities has different
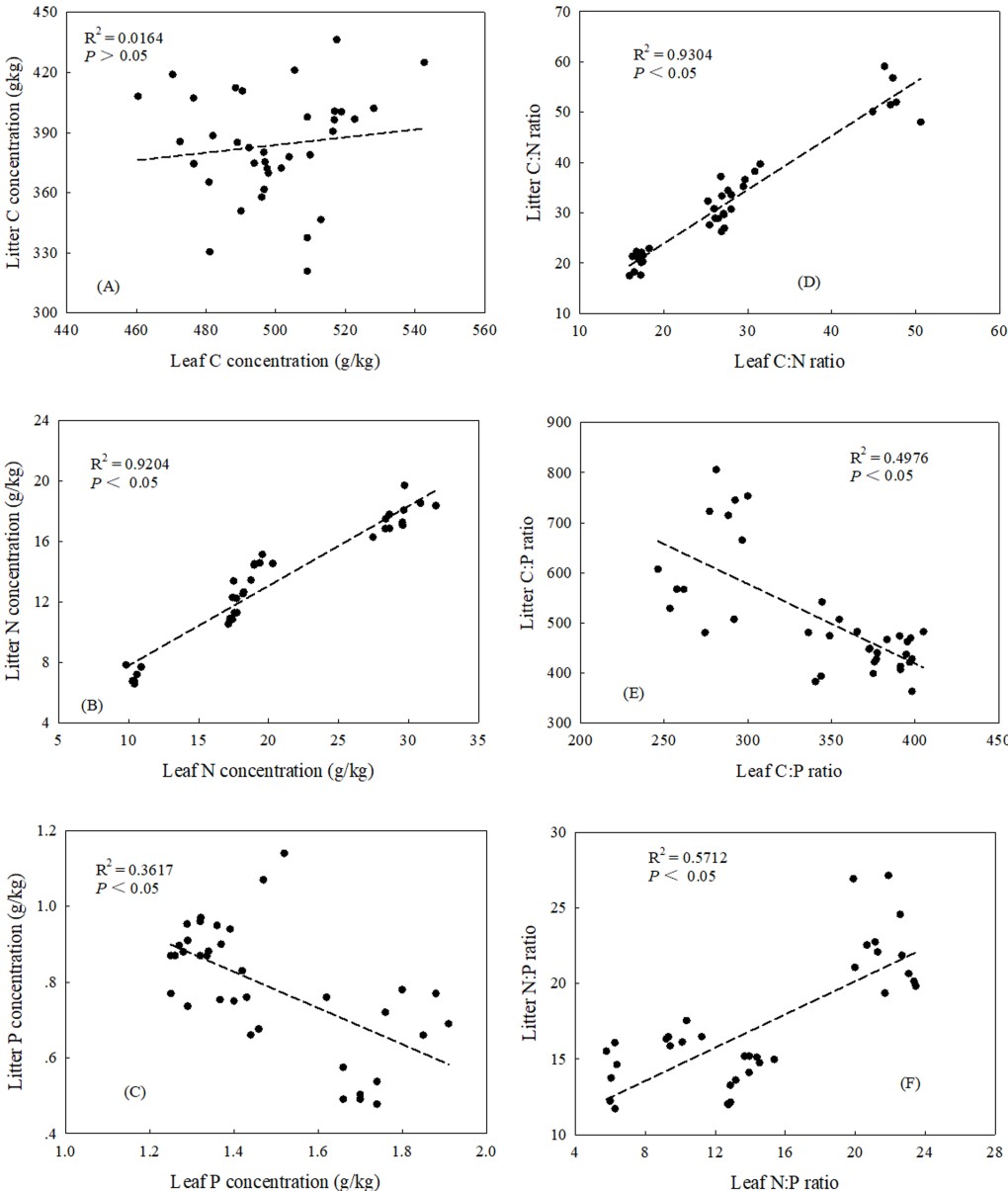

**Figure 2** **Relationships between leaf and litter C: N: P stoichiometric characteristics.** (A) The relationships between leaf and litter C concentrations. (B) The relationships between leaf and litter N concentrations. (C) The relationships between leaf and litter P concentrations. (D) The relationships between leaf and litter C:N ratios. (E) The relationships between leaf and litter C:P ratios. (F) The relationships between leaf and litter N:P ratios.

adaptability to the environment, and possess different strategies of nutrient adaptation (*Wright et al., 2004*; *Han et al., 2005*; *Zheng & Shangguan, 2007*; *He et al., 2008*; *Wu et al., 2012*). In this study, leaf C concentration in forest species was significantly higher than in grass and shrub species while the leaf P in forest species was significantly lower than in grass species. An explanation may be that trees construct nutrient poor woody tissues

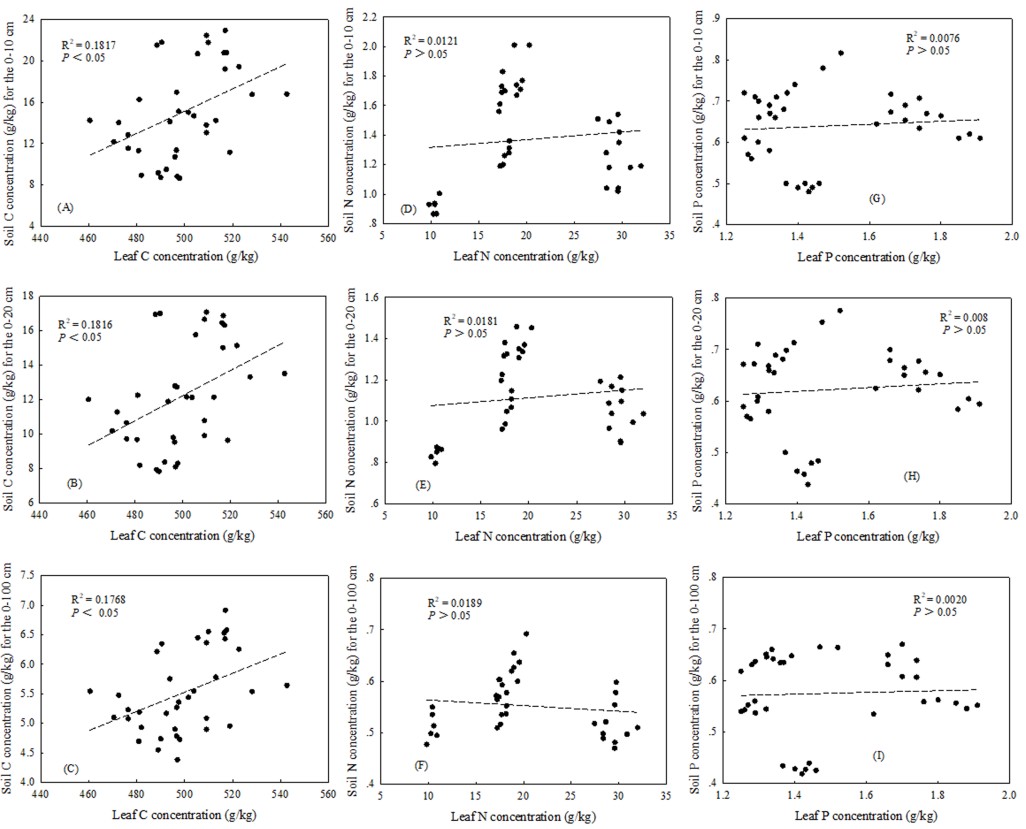

**Figure 3** **Relationships between leaf and soil C, N and P concentrations in 0–10/0–20 cm soil depths and 0–100 cm soil profile.** (A) The relationships between leaf and soil C concentrations in 0–10 soil depth. (B) The relationships between leaf and soil C concentrations in 0–20 soil depth. (D) The relationships between leaf and soil C concentrations in 0–100 cm soil profile. (D) The relationships between leaf and soil N concentrations in 0–10 soil depth. (E) The relationships between leaf and soil N concentrations in 0–20 soil depth. (F) The relationships between leaf and soil N concentrations in 0–100 cm soil profile. (G) The relationships between leaf and soil P concentrations in 0–10 soil depth. (H) The relationships between leaf and soil P concentrations in 0–20 soil depth. (I) The relationships between leaf and soil P concentrations in 0–100 cm soil profile.

while grasses do not. The results are consistent with those of *Wright et al. (2004)*, which reported that the leaf P concentration in herbaceous plants is significantly higher than in woody plants. Moreover, in this study, the C, N and P concentrations in plant leaves were higher than in the corresponding litter, which was consistent with previous studies (*Pan et al., 2011*; *Zeng et al., 2017*). *Pan et al. (2011)* showed that the C, N and P concentrations in the leaves of trees, shrubs and grasses were significantly higher than in litter, likely due to the reabsorption processes. Previous studies have shown that nutrients present in leaves are transferred to flowers, fruits, branches, and roots before leaf falling, thereby preventing nutrients loss (*Schreeg et al., 2014*). The results showed that N and P concentrations in litter varied greatly in different plant communities, and were significantly higher in trees than in grasses. This is because tree and shrub are deep-rooted plants, and have the strong

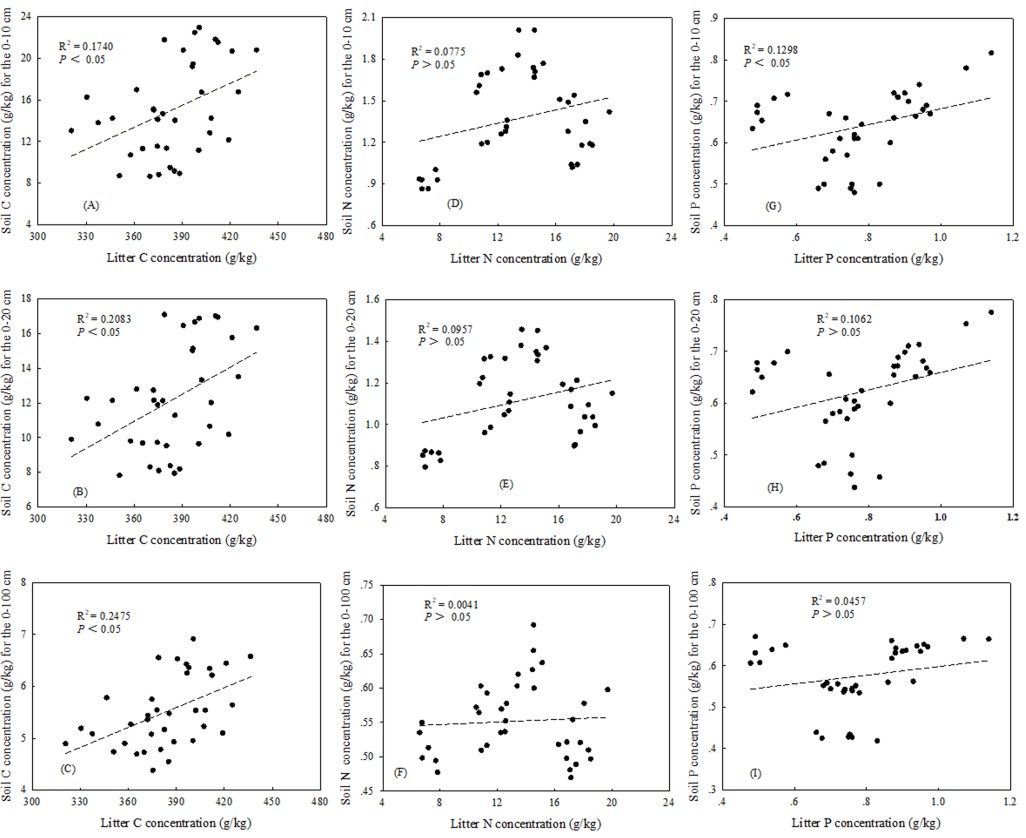

**Figure 4** **Relationships between litter and soil C, N and P concentrations in 0–10/0–20 cm soil depths and 0–100 cm soil profile.** (A) The relationships between litter and soil C concentrations in 0–10 soil depth. (D) The relationships between litter and soil C concentrations in 0–20 soil depth. (C) The relationships between litter and soil C concentrations in 0–100 cm soil profile. (D) The relationships between litter and soil N concentrations in 0–10 soil depth. (E) The relationships between litter and soil N concentrations in 0–20 soil depth. (F) The relationships between litter and soil N concentrations in 0–100 cm soil profile. (G) The relationships between litter and soil P concentrations in 0–10 soil depth. (H) The relationships between litter and soil P concentrations in 0–20 soil depth. (I) The relationships between litter and soil P concentrations in 0–100 cm soil profile.

capability of absorbing nutrients from multiple sources in the environment; while grasses have shallow roots and rely more on the recycling of their own nutrients.

N and P elements are major limiting factors for plant growth in terrestrial ecosystems, and the leaf N:P ratio could be used as an indicator to identify the limiting nutrient factors (*Koerselman & Meuleman, 1996*; *Güsewell, 2004*). However, the threshold of N:P ratio is affected by study area, plant growth stage and plant species (*Güsewell, 2004*). *Güsewell (2004)* reported that leaf N:P ratio can be used to reveal N-limitation (N:P ratio < 10) or P-limitation (N:P ratio > 20) in the ecosystem. In this study, based on the Güsewell's proposal that leaf N:P ratio indicates nutritional limitation for plant growth, we concluded that grass and shrub growth was limited by N and P element, respectively, whereas forest growth was co-limited by both of N and P elements together in the research area. In this study, the leaf N:P ratio in *S. viciifolia* and *H. rhamnoides* were 22.6 and 21.0, respectively,

**Table 6  Correlations among ecological stoichiometry in leaf, litter and soil in 0–10/0–20 cm soil depths and 0–100 cm soil profile.**

| Nutrient ratio | Soil depth (cm) | | | | | | | | |
|---|---|---|---|---|---|---|---|---|---|
| | **0–10** | | | **0–20** | | | **0–100** | | |
| | Soil C:N | Soil C:P | Soil N:P | Soil C:N | Soil C:P | Soil N:P | Soil C:N | Soil C:P | Soil N:P |
| Leaf C:N | −0.344[*] | −0.667[**] | −0.737[**] | −0.254 | −0.732[**] | −0.880[**] | −0.381[*] | −0.880[*] | −0.813[**] |
| Leaf C:P | 0.508[**] | 0.693[**] | 0.646[**] | 0.672[**] | 0.640[**] | 0.466[**] | 0.135 | 0.133 | −0.016 |
| Leaf N:P | 0.329 | 0.643[**] | 0.706[**] | 0.261 | 0.678[**] | 0.792[**] | 0.214 | 0.676[**] | 0.632[**] |
| Litter C:N | −0.395[*] | −0.708[**] | −0.758[**] | −0.303 | −0.762[**] | −0.886[**] | −0.305 | −0.874[**] | −0.809[*] |
| Litter C:P | −0.661[**] | −0.839[**] | −0.786[**] | −0.670[**] | −0.839[**] | −0.759[**] | −0.562[**] | −0.712[**] | −0.502[**] |
| Litter N:P | −0.092 | 0.183 | 0.313 | −0.273 | 0.258 | 0.542[**] | −0.108 | 0.591[**] | 0.721[**] |

**Notes.**
[*]Correlation is significant at the 0.05 level (2-tailed).
[**]Correlation is significant at the 0.01 level (2-tailed).

suggesting that shrub growth was P-limited. The leaf N:P ratios in *Q. liaotungensis* and *B. platyphylla* were 14.3 and 12.9, respectively, indicating that their growths were co-limited by both N and P. The leaf N:P ratios in *A. sacrorum* and *I. cylindrica* were 6.12 and 9.93, respectively, indicating that grass growth was limited by N. The results indicated that different plant communicates had different nutrient limiting elements, which was consisted with previous studies (*Han et al., 2005*). The reason is that grass species (*I. cylindrica* and *A. sacrorum*) is a shallow-rooted plant with a strong ability to absorb soil surface nutrients, particularly it has a greater capacity of relocating its leaf P before leaf falling than forest and shrub species, and it can more effectively utilize leaf P concentration to meet growth demands. Moreover, the biochemistry of the grass organic structure determines that more nitrogen is needed for growth. Therefore, grass species were less limited by P element than by N element. In addition, the results indicated that the growth of shrub species was limited by P element, which was similar to results reported by *Han et al. (2005)*. This is because *S. viciifolia* and *H. rhamnoides* are inherent species in vegetation restoration on the Loess Plateau and were nitrogen-fixing plants, and the absorption on of N element is far greater than that of P element, which results the shrub species to be limited by P element. Furthermore, the result showed that leaf C:N and C:P ratios were lower than in litter, which is consistent with results reported by *McGroddy, Daufresne & Heedin (2004)*, indicating that the reabsorption capacity for C is lower than for N and P. Although leaf N:P ratio can effectively reflect N or P limitation, the importance of the N:P ratio is mainly in its function as an indicator (*Güsewell, 2004*). If the leaf N:P ratio is to be used as an index to evaluate both N and P nutrient supplies in the Loess Plateau, further diagnosis regarding nutrient limitations should be conducted.

## Impacts of dominant plant communities on soil nutrient and ecological stoichiometry

Plants play an important role in improving soil fertility and contribute to the accumulation of soil nutrients. *Fu et al. (2010)* found that vegetation restoration could improve the net fixation of C and N and reduce their losses. However, the performance in soil quality

recovery differed among plant communities (*Jiao et al., 2012*; *Zeng et al., 2016*; *Deng et al., 2016*; *Zhao et al., 2017*). In this study, soil C, N and P concentrations in forestland was greater than in grassland and shrubland which is consistent with the previous results of *Jiao et al. (2012)* and *Qi et al. (2015)*. This result could be explained by a larger amounts of litter present in forestland, a more above-ground litter and a higher volume of root exudates reaching the soil, resulting in higher nutrient concentrations in the forestland than in other plant communities. Soil C and N concentrations decreased with increasing of soil depth, while soil P concentrations were relatively stable with depth, which was consistent with *Wei et al. (2009)*. The reasons might be the influence of soil parent material, the amount nutrient content of returning litter, the rate of decomposition, and plant nitrogen fixation, absorption and utilization. With an increasing of soil depth, the input of organic matter gradually decreased (*Nelson, Schoenau & Malhi, 2008*). However, soil P is mainly derived from rock weathering and leaching, and its mobility is very low, which caused vertical variation of P along the soil profile to be relatively stable (*Wei et al., 2009*).

Soil C:N:P ratios are important indicators of organic matter composition, soil quality and nutrient supply capacity (*Bui & Henderson, 2013*). In this study, soil C:N:P ratios among the three plant communities were 16.9:1.7:1, 25.0:2.3:1 and 28.6:2.5:1 at the topsoil (0–10 cm), respectively (Table 2), These values are substantially lower than the average global value of 186:13:1 (*Clevel & Liptzin, 2007*). Loess soils are naturally low in C, meanwhile, the Loess Plateau has undergone a serious soil erosion prior to recent efforts at vegetation restoration, resulting in a low C:N:P ratio. In this study, soil C:N ratio across different plant communities and soil depths was approximately 10.8 in the Loess Plateau, which was similar to the average level (11.9) in China (*Tian et al., 2010*), but lower than the world's average value of 13.3 (*Clevel & Liptzin, 2007*). Previous studies showed that soil C:N ratio is negatively correlated with the decomposition rate of organic matter, and low soil C:N ratio indicates that organic matter is well decomposed (*Zhao et al., 2015*; *Deng et al., 2016*). The soil C:N ratio in grassland, shrubland and forestland was 10.1, 10.8 and 11.7, respectively, implying that organic matter had been completely decomposed. The soil C:N ratio in each plant community maintained relative stability with increasing soil depth, which is consistent with previous studies (*Tian et al., 2010*). This may be due to the same change dynamics in C and N. Soil C:P and N:P ratios in each plant community decreased with increasing soil depth, which may be due to the difference in the source of soil C, N and P. Furthermore, this study showed that soil C:P and N:P ratios in forestland was higher than in shrubland and grassland in the topsoil depth, which may be due to the fact that forest had more above-ground biomass than shrubland and grassland (*Qi et al., 2015*).

## Relationships between C, N and P concentrations and their characteristics of ecological stoichiometry among leaf, litter and soil

Some previous studies have showed a strong correlation between leaf and soil nutrients (*Parfitt, Yeates & Ross, 2005*; *Agren & Bosatta, 1998*; *Agren, 2008*), while others found that there was no correlation between N and P concentrations in leaf and soil (*Ladanai, Agren & Olsson, 2010*; *Yu et al., 2010*). In this study, no significant correlation was found between soil N concentration with leaf N concentration for three plant community types. One

possible reason is that through long-term adaptation to the habitat, the N concentration in plant leaves in this region may be more affected by the attributes of the species than the limitation of soil nutrients. In addition, *Reich & Oleksyn (2004)* showed that the mineral elements of plants are a combination of climate, soil nutrients and species composition. Other studies have suggested that soil temperature, soil water concentration, microbial activity and other factors have a greater impact on the mineral elements of plants (*Chapin & Pastor, 1995*; *Güsewell, 2004*). In this study, there was a significant correlation between litter N and P concentrations and their ratios with leaf N and P concentrations among the three plant types, indicating that the nutrients in litter were derived from plant leaves. In addition, a strong correlation between soil and litter for both C and P concentrations among the three plant types was observed. As a considerable portion of C and other nutrients elements in the litter could be released into the soil, such that litter was one of the main sources of soil nutrients (*Agren et al., 2013*). In general, this study showed that there is a close correlation between the concentrations of C, N and P and their ratios in leaf, litter and soil in three plant community types, which confirmed that C, N and P in the ecosystem were transported and transformed among plants, litter and soil (*McGroddy, Daufresne & Heedin, 2004*).

## CONCLUSION

This study analyzed C, N and P concentrations and their stoichiometric characteristics in leaf, litter and soil of three dominant plant types: grass (*I. cylindrica* and *A. sacrorum*)), shrubs (*S. viciifolia* and *H. rhamnoides*) and tree (*Q. liaotungensis* and *B. platyphylla*)) during vegetation restoration on the Loess Plateau of China. The results indicated that plant community type had significant effects on leaf, litter and soil nutrient concentrations, and their stoichiometry characteristics. Grass species had highest leaf P concentration and forest species litter had highest P concentration. Leaf C, N and P concentrations were higher than in litter and soil ($P < 0.05$) and forest community type had highest soil nutrient concentrations at all soil layers and their ecological stoichiometries were highest in topsoil ($P < 0.05$). In addition, soil C:N:P ratios in all plant communities decreased with increasing soil depth. Soil P concentration and N:P ratio had significant positive correlations with litter P concentration and N:P ratio for the three plant community types ($P < 0.05$), respectively. However, there were no significant correlations between soil N, P concentrations and N:P ratio with leaf N and P concentrations and N:P ratio ($P > 0.5$), respectively. Based on the theory that leaf N:P ratio indicates nutritional limitation for plant growth, this study concluded that plant growth of the forest community type (*Q. liaotungensis* and *B. platyphylla* species) was co-limited by both of N and P elements, plant growth of shrub community type (*H. rhamnoides* and *S. viciifolia* species) was limited by P element and grass growth (*I. cylindrica* and *A. sacrorum* species) was limited by N element. These results can provide a scientific basis for the reconstruction of degraded ecosystem on the Loess Plateau of China.

## ACKNOWLEDGEMENTS

The authors are indebted to the anonymous reviewers and editors for their thoughtful comments and valuable suggestions and the authors also thank Dr. Glenn V. Wilson for correcting the English.

### Funding

This study was financially supported by the External Cooperation Program of Chinese Academy of Sciences (Grant No. 161461KYSB20170013). The funders had no role in study design, data collection and analysis, decision to publish, or preparation of the manuscript.

### Grant Disclosures

The following grant information was disclosed by the authors:
External Cooperation Program of Chinese Academy of Sciences: 161461KYSB20170013.

### Competing Interests

The authors declare there are no competing interests.

### Author Contributions

- Zongfei Wang conceived and designed the experiments, performed the experiments, analyzed the data, prepared figures and/or tables, authored or reviewed drafts of the paper, and approved the final draft.
- Fenli Zheng conceived and designed the experiments, performed the experiments, authored or reviewed drafts of the paper, and approved the final draft.

### Data Availability

  The raw measurements are available in File S1.

### Supplemental Information

Supplemental information for this article can be found online at http://dx.doi.org/10.7717/peerj.10084#supplemental-information.

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
