# Peer review of "Ecological stoichiometry of plant leaves, litter and soils in a secondary forest on China’s Loess Plateau"

_PeerJ, doi:10.7717/peerj.10084_

## Round 0.1 · original submission · Major Revisions

Overall the reviewers comments are fairly similar to each other, and I agree with the assessments in both cases. This manuscript has potential, but it has a long way to go to reach publishable standard. Several fundamental things need addressing first; including the logical flow of arguments, the repetition within the manuscript, clearer methodological description, correct statistical analysis and vastly improved English language. In short, every detail mentioned by your reviewers needs addressing, along with a clear explanation in your response letter.

Reviewer 1 ·

Basic reporting

see "General comments for the author"

Experimental design

see "General comments for the author"

Validity of the findings

see "General comments for the author"

Additional comments

The present ms. studied the C, N and P concentrations and their ratios in leaves, litter and soils in a temperate forest.
The authors found that all these parameters varied among plant species and soil profiles. They also found that plant C/N/P were generally not correlated to soil nutrients.

Although the ms. is clear to follow and formulated, in my view the ms. in its present form suffers from some problems that should be thoroughly taken into account before acceptation.

My first comment is concerning the language, which needs to be substantially improved. I had problems understanding many statements. In many places, the ms. is poorly written in English – or what is otherwise known as Chinglish. For example, the wording ‘characteristics’ and 'area' in the title, if considering language only, shall be removed. The wording ‘in dominant species of trees, bushes and grasses’ shall be also removed. The title may change to 'Ecological stoichiometry of plant leaves, litter and soils in a secondary forest on China's Loess Plateau'.

In the first opening sentence in the section of Abstract, ‘…interactions between soil, plants and their nutrient cycle’, although it is grammatically correct, this kind of English expression is weird and unusual. Also, in the 2nd sentence, ‘there is litter information…’ shall be corrected. And more Chinglish, even grammar errors, could be found in this submission. I highly recommend that the manuscript is proof read by a native English speaking person before the reviewer process proceeds. The grammar could be improved in many places through the manuscript. It is, however, not only a matter of grammar; the ideas are sometimes difficult to follow, particularly in the Discussion section, where it is difficult to grasp what the authors mean because of the way data are presented.

Introduction: The introduction presents a number of ideas but lacks organization. Consequently, it is difficult to follow and pull out the justification and intent of the research.

The 1st paragraph talks about soil/ecosystem erosion, degradation and restoration. However, the data presented in this ms are not strongly associated to these terms and they are not adequately discussed in the section of Discussion.

M&M: The methods do not provide details of how the “six plant species” were selected. How many species are in the sites? Why only select 6 dominant species? Due to so many species in your sites, using averaging data of only 2 tree/shrub/grass species will lead to different results of statistical analysis and thus the discussion of growth form differences is largely speculative given the information presented.

Statistical analysis appears to be incorrect, or at least poorly justified. The authors need to present their statistical results of 2-way ANOVA. The authors must have misunderstood correlation and regression analysis (Table 5).

Three digitals are enough when reporting numbers.

All letters of R, P and n that are used to represent R square, probability and sample size shall be changed to be in italic.

Results: There are far too many graphs. I do not believe they are all necessary. Please think better about the exact message you want to make. I do not think you need to plot all leaf/litter/soil parameters against each other. Some pictures seem to tell the same story. Just present your most important points and some pictures can be presented in tables or moved to supplementary materials.

Figure 1 is repetitive compared to Table 5.

Discussion: I found the Discussion to repeat many of the concepts or arguments presented earlier and the logic often not particularly compelling. Overall, I felt this was the weakest section of the paper. Some of the statements in the Discussion are very speculative. Sometimes the authors seem to be making the same point but just in a slightly different way and they are urged to keep to the key issues. While it is important to compare results with other work, the discussion seems to compare results with all other work individually and in great detail, rather than giving a general overview of how this work fits in with previous findings (i.e. it reads too much like the first draft of a Ph.D. thesis in places!).

·

Basic reporting

This paper reports on an observational study of nutrient concentration and C:N:P stoichometry in three different plant communities growing on the loess plateau in china. All three communities may have originated from systematic ecosystem reconstruction initiated after severe soil erosion ensued post-deforestation. The paper showcases all differences between nutrient concentrations and stoichiometric ratios in fresh foliage, litter and various soil layers up to 1 m depth. The authors conclude that these tree communities – grassland, shrubland and forest – do indeed differ in plant tissue stoichiometry and this is reflected in the soil. The paper is fairly well written, although it would benefit from editorial review, there are some examples of potential improvement in the attached pdf.

This study is potentially interesting, however it needs substantial improvement if it is to be considered for publication. The main issues are lack of originality and poor description of methodology. These two issues must be addressed before the rest of the paper can be reviewed. For example, authors find that ‘soil sampling depth had an effect on nutrient concentration’ – this is not a novel finding, I believe we have known this for some time. Similarly, the paper would benefit from a much better description of the studies plant species and their community – species identity is one of the main drivers or its relation to the environment, including the nutrient cycle. In the discussion, the authors posit that ‘This is because shrub species was nitrogen-fixing plants, and their absorptions of N element are much greater than that the P element, resulting in a lack of P element in shrub specie.’ Well, obviously, but this information needs to be presented in Methodology so that the reader knows what to expect from each plant type.

Experimental design

This is an observational study analysing a single observation period, which is fine for a direct comparison of differences among plant types, but little further inference as to the ecology of these communities can be drawn. The sampling design is poorly described, it is not clear what is the replication rate. From the description available, it may well be that the underlying rate of replication is 1, severely reducing the scope of statistical analyses. It is not clear how many soil samples were collected, the authors mention an S-shaped sampling line, but no further detail is provided. Analytical methods are adequate for the task, however the use of a fairly strong acid to extract soil P is likely to provide an estimate of total P, most of which is not available to plants.

Validity of the findings

The findings and data reported in the paper are within expected range and probably describe the snapshot concentrations and ratios found on the day of the sampling fairly well. Beyond that, it is very difficult to assess the validity of the comparisons, mainly due to insufficient information on field sampling designs and on statistical analyses.

Additional comments

The authors need to improve their presentation of the objectives and the motivation for the paper, clarify methodology and statistical approach, shorten the results section (there is a lot of repetition between the text and the tables/figures), reduce the amount of figures by 2/3 and improve their clarity, reduce the length of the discussion and focus it on key findings only.

---

## Round 0.2 · Minor Revisions

Please pay particular attention to the annotated manuscript as well as the reviewer comments.

·

Basic reporting

The presentation, layout of the document and language flow are much improved. The main message of the paper is now clearly visible.

Experimental design

Much better sampling design description, it is now possible to reproduce the study on the basis of the information presented in the paper.

Validity of the findings

Most of the findings presented in the paper are confirmatory, however stoichiometry of plant-soil systems is a developing field where additional observations from unusual systems are necessary to build a global picture.

Additional comments

A much improved version of the paper, I would like to thank the authors for taking our comments on board. There are some small editorial suggestions in the attached pdf, I am happy to recommend acceptance.

---

## Round 0.3 · accepted · Accept

Thank you for dealing with the reviewers' comments so thoroughly. Your paper will make a useful contribution to our understanding of the ecological stoichiometry of nutrients in secondary forests.